# Correlation between Component Factors of Non-Alcoholic Fatty Liver Disease and Metabolic Syndrome in Nurses: An Observational and Cross-Sectional Study

**DOI:** 10.3390/ijerph192316294

**Published:** 2022-12-05

**Authors:** Wen-Pei Chang, Yu-Pei Chang

**Affiliations:** 1Department of Nursing, Shuang Ho Hospital, Taipei Medical University, New Taipei City 235, Taiwan; 2School of Nursing, College of Nursing, Taipei Medical University, Taipei 110, Taiwan

**Keywords:** nurse, non-alcoholic fatty liver disease, metabolic syndrome

## Abstract

This study aimed to understand the correlation between non-alcoholic fatty liver disease (NAFLD) and metabolic syndrome in nurses. Questionnaires were used to eliminate individuals with a daily drinking habit, hepatitis B or C, or incomplete data. A total of 706 valid samples were obtained. The prevalence of NAFLD among nurses was 36.8%. Nurses with a greater age (OR = 1.08, 95% CI: 1.01–1.16), obese BMI (OR = 23.30, 95% CI: 8.88–61.10), overweight BMI (OR = 3.89, 95% CI: 2.15–7.04), waist circumference exceeding the standard (OR = 2.10, 95% CI: 1.14–3.87), fasting blood glucose 100–125 mg/dL (OR = 4.09, 95% CI: 1.19–14.03), and overly low HDL-C (OR = 2.01, 95% CI: 1.05–3.85) were at greater risk of NAFLD. Furthermore, male nurses (OR = 6.42, 95% CI: 1.07–38.70), nurses with triglycerides over 150 mg/dL (OR = 4.80; 95% CI: 1.05–21.95), and nurses with HDL-C lower than the standard (OR = 5.63, 95% CI: 1.35–23.49) were at greater risk of moderate/severe NAFLD. Among younger nurses, those of greater age, male nurses, obese and overweight nurses, and those with a waist circumference exceeding the standard, 100–125 mg/dL, overly low HDL-C, and triglycerides over 150 mg/dL should consider the possibility that they have NAFLD.

## 1. Introduction

Although the healthcare knowledge, health beliefs, and access to healthcare of nurses are superior to those of the general public, nurses are not necessarily healthier [1]. Nursing is an occupation with high risk, high stress, and long working hours. Shift work and the stress and demands of clinical work may force nurses to change their daily habits, thereby increasing the risk of hypertension, cardiovascular diseases, and obesity among nurses. For this reason, nurses are at a higher risk of developing metabolic syndrome [2].

Non-alcoholic fatty liver disease (NAFLD) is a collective term for diseases spanning from simple steatosis and steatohepatitis to cirrhosis [3]. In recent years, this group of diseases has begun to receive more attention due to its prevalence among the general population. NAFLD is currently the most common type of liver disease around the world and has the potential to develop into end-stage liver disease [4]. A diagnosis of NAFLD is defined as the presence of imaging or histology evidence of hepatic steatosis when other causes of fatty liver, such as heavy alcohol use, drugs, and hereditary diseases, have been eliminated [5]. At present, there is no established therapy or drug for NAFLD, and treatment is limited to regular exercise and weight control [6].

As the significance of metabolic syndromes has become increasingly apparent, relevant studies have indicated that NAFLD, which is associated with metabolic syndrome, is of clinical importance [7,8]. Not only is metabolic syndrome closely linked to common chronic diseases such as cerebrovascular disease, heart disease, diabetes, and hypertension, but it can also result in NAFLD. Common comorbidities of NAFLD include obesity, hypertension, diabetes, and hyperlipidemia, and NAFLD is also associated with cardiovascular diseases and insulin resistance. Some researchers even believe that NAFLD is the manifestation of metabolic syndrome in the liver [9].

According to Caruso et al., nurses in a better state of health can provide patients with better quality healthcare services [10]. The unique attributes of nursing work mean that the health of nurses influences patients. In view of the uniqueness of the work of hospital personnel and the importance of their health conditions and with the increasing severity of NAFLD, metabolism syndrome, and obesity, NAFLD in nurses deserves attention; however, few studies have investigated the relevant factors of this group. This study, therefore, performed an analysis of this group to provide a reference to promote the health of nurses. We proposed hypotheses and established a research framework (Figure 1):

**Hypothesis** **1** **(H1).***The development of NAFLD among nurses varies significantly according to personal background variables (including age, gender, marital status, education, years of service, drinking habits, smoking habits, and shift work conditions)*.

**Hypothesis** **2** **(H2).***The development of NAFLD among nurses varies significantly according to physical indexes (including BMI, waist circumference, hypertension, fasting blood glucose, triglycerides, total cholesterol, and HDL-C)*.

**Figure 1 ijerph-19-16294-f001:**
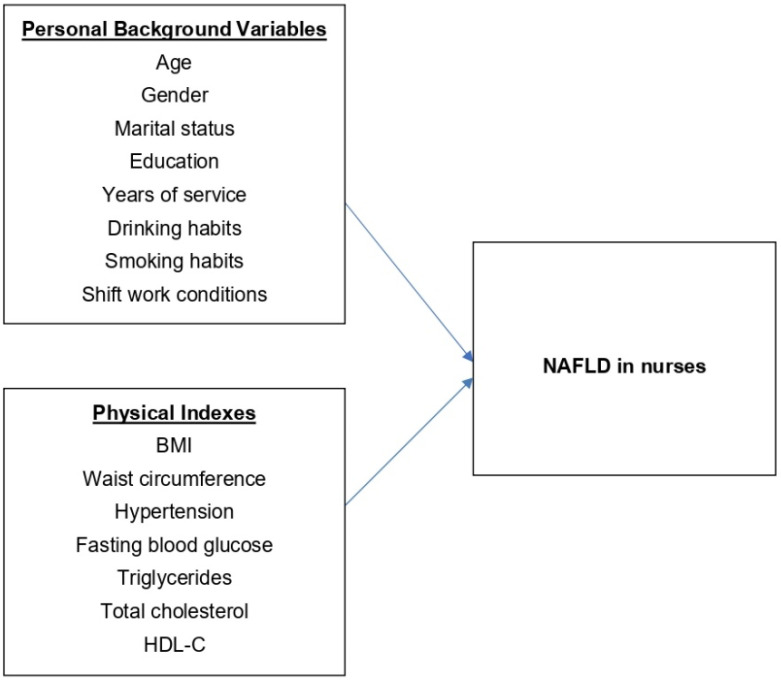
Research framework of factors influencing the incidence of NAFLD in nurses.

## 2. Materials and Methods

### 2.1. Participants and Sampling

After gaining approval from the institutional review board, we used data from a routine annual health checkup for nurses at a teaching hospital in northern Taiwan. The health checkup results were collected between February 2018 and April 2018. After removing any personally identifiable information, we used questionnaires to eliminate individuals with a daily drinking habit, hepatitis B or C, or incomplete data. A total of 706 valid samples were obtained.

### 2.2. Design

Nurses in this study were assessed their drinking habits by choosing an answer from the three choices: non-drinking, drinking alcohol occasionally (in social situations), and drinking alcohol every day. Those who chose ‘drinking alcohol every day’ were not allowed to participate in this study (only one person clicked this choice and was excluded from the study). Those with hepatitis B or C were eliminated via medical history queries or records of clinical visits to the hospital in the past (there were five individuals with hepatitis B and two individuals with hepatitis C). For marital status, those who circled ‘widowed’, ‘single’, ‘separated’, or ‘divorced’ were categorized as not living with a spouse or partner, and those who circled ‘married’ or ‘cohabiting’ were categorized as living with a spouse or partner. For smoking habits, those who circled ‘never’ were categorized as having no smoking habits. For shift type, only those who circled ‘long-term rotating shifts’ or ‘night shifts’ were categorized as shift work; anything else was categorized as non-shift work. In accordance with the metabolic syndrome definition revised by the Health Promotion Administration, of the Ministry of Health and Welfare [11] in Taiwan and the National Cholesterol Education Program Adult Treatment Panel III guidelines (National Cholesterol Education Program (NCEP) Expert Panel on Detection, Evaluation, and Treatment of High Blood Cholesterol in Adults (Adult Treatment Panel III) [12], obesity was defined using the body mass index (BMI), which equals body weight (kg)/height^2^ (m)^2^. BMI < 24, 27 > BMI ≥ 24, and BMI ≥ 27 were defined as normal, overweight, and obese, respectively. The demarcation points of waist circumference for men and women were 90 cm and 80 cm. The demarcation point of blood pressure was 130/85 mmHg and a hypertension medical history. The demarcation point between normal and fasting blood glucose was 100 mg/dL; fasting blood glucose between 100 mg/dL and 125 mg/dL was regarded as impaired fasting glucose (IFG) and blood glucose over 126 mg/dL was regarded as diabetes. Triglyceride levels higher than 150 mg/dL were deemed to indicate hypertriglyceridemia. Blood cholesterol greater than or equal to 200 mg/dL was considered to indicate hypercholesterolemia. High-density lipoprotein cholesterol (HDL-C) levels were required to be greater than or equal to 40 mg/dL in men and greater than or equal to 50 mg/dL in women.

Fatty livers were diagnosed based on abdominal ultrasounds, with the executing physician being a hepatobiliary specialist. Based on ultrasound imaging characteristics, the diagnoses were divided into three grades: no fatty liver, mild fatty liver (significantly increased echogenicity on the liver surface compared to the kidney but clear imaging of intrahepatic vessels), and moderate/severe fatty liver (the same imaging characteristics as mild fatty liver but unclear imaging of intrahepatic vessels and deep hepatic parenchyma obscuring) [13].

### 2.3. Statistical Methods

Data analysis was performed using SPSS 22 (SPSS Inc, Chicago, IL, USA). The statistical methods included descriptive analysis to examine the demographic and biochemical variable distributions and chi-square tests and t-tests to compare the demographics, relevant risk factors, and various laboratory test data of nurses with NAFLD and those with no fatty liver.

To identify the primary factors that influence the severity of NAFLD in nurses, we conducted a multinomial logistic regression analysis and determined whether various personal background variables (including age, gender, marital status, educational background, years of service, drinking habits, smoking habits, and shift work conditions) and physical indexes (including BMI, waist circumference, hypertension, fasting blood glucose, triglycerides, total cholesterol, and HDL-C) were risk factors of NAFLD in nurses. We obtained their odds ratios (Ors) and 95% confidence intervals (Cis).

## 3. Results

### 3.1. Prevalence of NAFLD among Nurses and Basic Information Analysis

This study collected a total of 706 valid samples, among which 30 (4.2%) were men and 676 (95.8%) were women. The average age was 29.39 ± 7.06 and the overall prevalence rate of NAFLD was 36.8% (260 nurses), of which 26.6% (188 nurses) had mild NAFLD and 10.2%) (72 nurses) had moderate/severe NAFLD. Furthermore, nurses with NAFLD tended to be older (*p* < 0.001), have worked for longer (*p* < 0.001), did less shift work (*p* = 0.037), have hypertension (*p* = 0.031), and have a higher BMI (*p* < 0.001), waist circumference (*p* < 0.001), fasting blood glucose (*p* < 0.001), triglycerides (*p* = 0.001), and HDL-C (*p* < 0.001) than those with no NAFLD. The differences were statistically significant (Table 1).

### 3.2. Primary Influencing Factors of NAFLD in Nurses

Table 2 presents the relevant risk factors of NAFLD in nurses. The results of our multinomial logistic regression analysis revealed that the factors with significant regression coefficients included age (OR = 1.07, 95% CI: 1.00–1.15), BMI obese vs. normal (OR = 38.29, 95% CI: 14.74–99.46) and overweight vs. normal (OR = 4.12, 95% CI: 2.31–7.32), waist circumference exceeding the standard vs. normal (OR = 2.27, 95% CI: 1.25–4.14), fasting blood glucose 100–125 mg/dL vs. <100 mg/dL (OR = 4.66, 95% CI: 1.42–15.26), and HDL-C overly low vs. normal (OR = 2.01, 95% CI: 1.11–3.97).

### 3.3. Comparison of Primary Factors Influencing the Severity of NAFLD in Nurses

The results from the multinomial logistic regression analysis in Table 3 indicate that compared with nurses without NAFLD, the factors with significant regression coefficients in nurses with mild NAFLD include age (OR = 1.08, 95% CI: 1.01–1.16), BMI obese vs. normal (OR = 23.30, 95% CI: 8.88–61.10) and overweight vs. normal (OR = 3.89, 95% CI: 2.15–7.04), waist circumference exceeding the standard vs. normal (OR = 2.10, 95% CI: 1.14–3.87), fasting blood glucose 100–125 mg/dL vs. <100 mg/dL (OR = 4.09, 95% CI: 1.19–14.03), and HDL-C overly low vs. normal (OR = 2.01, 95% CI: 1.05–3.85).

When nurses with moderate/severe NAFLD were compared with those without NAFLD, the factors with significant regression coefficients included gender male vs. female (OR = 6.42, 95% CI: 1.07–38.70), BMI obese vs. normal (OR = 409.63, 95% CI: 61.59–2724.38) and overweight vs. normal (OR = 13.50, 95% CI: 2.35–77.50), waist circumference exceeding the standard vs. normal (OR = 9.33, 95% CI: 2.30–37.98), fasting blood glucose 100–125 mg/dL vs. <100 mg/dL (OR = 11.74, 95% CI: 2.22–62.27), triglycerides ≥150 mg/dL vs. <150 mg/dL (OR = 5.63, 95% CI: 1.35–23.49), and HDL-C overly low vs. normal (OR = 2.60, 95% CI: 1.03–6.58) (Table 3).

## 4. Discussion

This study found that the prevalence of NAFLD among nurses was 36.8%, which is close to the results of other studies. The prevalence of NAFLD among adults ranges from 15% to 30% in western countries and from 15% to 45% in Asian regions such as Southeast Asia, Korea, Japan, and Taiwan [14,15,16].

Animal testing has revealed that chronically disrupted circadian rhythms can cause NAFLD and, in turn, lead to liver cancer. Like other organs in the human body, the liver follows a 24-h circadian rhythm, and mealtimes are believed to be what reminds the liver to follow a 24-h cycle [17]. Yasutake et al., maintained that regular mealtimes are sufficient to reduce the risk of NAFLD [16]. More specifically, one should not skip meals or eat late at night. However, this study only discussed the correlation between mealtimes and NAFLD; more specific or potential pathogenic mechanisms will require further research to clarify. This study did not find that doing shift work increases the risk of NAFLD in nurses. However, it may be that the nurses examined in this study were younger and had done fewer years of service, which prevented us from proving this.

The ages of the nurses with NAFLD were significantly higher than those without NAFLD, which proves that NAFLD is more likely to occur with age. Similar results have been found in other previous studies [18,19,20,21]. Sayiner et al., even claimed that age is an independent predictor of NAFLD severity [21]. Past studies have also found a higher prevalence of obesity and metabolic syndrome in men than in women [22,23] and that NAFLD is influenced by metabolism-related conditions such as weight, waist circumference, blood glucose, and blood lipids [24,25,26]. Animal studies have shown that the androgen/androgen receptor pathway specific to males may be a key risk factor in the pathogenesis of NAFLD [27]. Our results similarly presented a higher risk of NAFLD in male nurses than in female nurses; thus, this gender tendency is consistent with that found in foreign and domestic papers previously published.

In Taiwan, economic progress and the westernization of eating habits in recent years have caused increases in obesity, diabetes, and hyperlipidemia among the population. Yeh et al. investigated 6790 hospital workers and found a metabolic syndrome prevalence rate of 12.0% [28]. Schwenger and Allard suggested that the prevalence rate of NAFLD is high at 58% among overweight adults and extremely high at 98% among obese adults, which shows that NAFLD is indeed associated with overweight and obesity [16]. This study also found NAFLD to be correlated with overweight and obesity in nurses; however, the prevalence rate of NAFLD was 20.8% among overweight nurses and 48.5% among obese nurses, which is lower than that found in previous research. This may be because most of the nurses in this study were women and younger.

The pathogenic mechanisms of NAFLD are fairly complex. Abnormal fat metabolism, the generation of reactive oxygen species, increased lipid peroxidation in hepatocytes, activated stellate cells, and abnormal cytokine production can all cause hepatocyte damage and fibrosis. Thus, steatosis in the liver may develop into NAFLD [29,30]. Although Lin et al. predicted that the risk factors of NAFLD in Taiwan would include male gender, high age, high BMI, high alanine aminotransferase (ALT), high triglycerides, and high total cholesterol, this study found that the high-risk factors of NAFLD were overweight and obese BMI, waist circumference exceeding the standard, fasting blood glucose 100–125 mg/dL, and overly low HDL-C [31].

Other main causes of metabolic syndrome are obesity and insulin resistance. Insulin resistance means that cells cannot effectively utilize insulin, which means that they cannot properly turn glucose into energy. The free fatty acids in the blood then enter the liver and accumulate in liver cells in the form of fat. Endotoxins, cytokines, oxidant production, and oxidative stress in the body then result in the inflammation and fibrosis of the liver. Both metabolic syndrome and NAFLD are mainly caused by obesity and insulin resistance, and metabolic syndrome is also associated with NAFLD severity. A very close relationship thus exists between the two [32,33]. The above also supports another finding of this study as, aside from overweight and obese BMI, waist circumference exceeding the standard, and fasting blood glucose 100–125 mg/dL being possible high-risk factors of NAFLD, nurses with triglycerides over 150 mg/dL and HDL-C lower than the standard are at greater risk of moderate/severe NAFLD than nurses with triglycerides below 150 mg/dL and normal HDL-C.

Preventive healthcare for nurses warrants attention. NAFLD is a chronic disease and, as it progresses, it increases the chance of cirrhosis, liver cancer, diabetes, and cardiovascular diseases, thereby threatening the health of nurses. NAFLD and metabolic syndrome may be consequences of each other. The best way to break this vicious cycle would be to control the weight, fasting blood glucose, triglycerides, and HDL-C of nurses [34].

## 5. Conclusions

In conclusion, the close relationship between metabolic syndrome and NAFLD is significant to nurse health. This study found that the prevalence rate of NAFLD among nurses was 36.8% and that nurses with a greater age, overweight and obese BMI, waist circumference exceeding the standard, fasting blood glucose 100–125 mg/dL, and overly low HDL-C were at greater risk of NAFLD. Male nurses and nurses with triglycerides over 150 mg/dL were at greater risk of moderate/severe NAFLD than female nurses and nurses with triglycerides below 150 mg/dL. Thus, nurses should adopt relevant strategies. Self-management and proper control of weight, blood glucose, and HDL-C are recommended for older nurses to prevent NAFLD. Regular blood tests and abdominal ultrasounds are recommended for male nurses and nurses with higher triglycerides to prevent moderate or severe NAFLD. The results of this study can provide directions for future research, including obtaining observations on whether lifestyle changes or exercise interventions can prevent or control the incidence of NAFLD in nurses.

## 6. Study Limitations

The samples in this study were limited to nurses who had undergone health checkups. Therefore, we were unable to collect more samples. Furthermore, of the 706 samples, only 30 were men, with the remainder all being women, which may have affected the interpretations of this study. We also did not include the department in which the nurses worked or their living or dietary habits (including whether any drugs and/or herbal supplements were being taken); thus, these variables could not be controlled. Furthermore, this was a cross-sectional study; the results show which risk factors NAFLD is associated with but do not confirm which factors can be used to predict the occurrence of NAFLD. A prospective study of the nurses who do not have NAFLD would deal with timing issues and confirm the relevant risk factors of NAFLD.

## Figures and Tables

**Table 1 ijerph-19-16294-t001:** Analysis of NAFLD and basic information of nurses.

Variable	All *n* = 706	Without NAFLD *n* = 446	With NAFLD *n* = 260	*p*
	No. (%)	No. (%)	No. (%)	
Gender				0.713
Male	30 (4.2)	18 (4.0)	12 (4.6)	
Female	676 (95.8)	428 (96.0)	248 (95.4)	
Marital status				0.564
Living with spouse or partner	136 (19.3)	83 (18.6)	53 (20.4)	
Not living with spouse or partner	570 (80.7)	363 (81.4)	207 (79.6)	
Education				0.910
Junior college	250 (35.4)	155 (34.8)	95 (36.6)	
University	420 (59.5)	272 (61.0)	148 (56.9)	
Graduate school	36 (5.1)	19 (4.2)	17 (6.5)	
Drinking				0.559
Yes (including occasionally or only when entertaining)	286 (40.5)	177 (39.7)	109 (41.9)	
No	420 (59.5)	269 (60.3)	151 (58.1)	
Smoking				0.503
Yes	32 (4.5)	22 (4.9)	10 (3.8)	
No	674 (95.5)	424 (95.1)	250 (96.2)	
Shift work				0.037
Yes	531 (75.2)	347 (77.8)	184 (70.8)	
No	175 (24.8)	99 (22.2)	76 (29.2)	
BMI				<0.001
Normal	478 (67.7)	398 (89.2)	80 (30.8)	
Overweight	95 (13.5)	41 (9.2)	54 (20.8)	
Obese	133 (18.8)	7 (1.6)	126 (48.5)	
Waist circumference				<0.001
Normal	171 (69.3)	396 (88.8)	93 (35.8)	
Exceeding standard	217 (30.7)	50 (11.2)	167 (64.2)	
Hypertension				0.031
Yes	16 (2.3)	6 (1.3)	10 (3.8)	
No	690 (97.7)	440 (98.7)	250 (96.2)	
Fasting blood glucose (mg/dL)				<0.001
<100	669 (94.8)	438 (98.2)	231 (88.8)	
100–125	26 (3.7)	6 (1.3)	20 (7.7)	
≥126	11 (1.6)	2 (0.4)	9 (3.5)	
Triglycerides (mg/dL)				0.001
<150	670 (94.9)	433 (97.1)	237 (91.2)	
≥150	36 (5.1)	13 (2.9)	23 (8.8)	
Total cholesterol (mg/dL)				0.131
<200	507 (71.8)	329 (73.8)	178 (68.5)	
≥200	199 (28.2)	117 (26.2)	82 (31.5)	
HDL-C (mg/dL)				<0.001
Normal	597 (84.6)	411 (92.2)	186 (71.5)	
Overly low	109 (15.4)	35 (7.8)	74 (28.5)	
NAFLD				
No	446 (63.2)	-	-	
Mild	188 (26.6)	-	-	
Moderate/severe	72 (10.2)	-	-	
	Mean (*SD*)	Mean (*SD*)	Mean (*SD*)	
Age (years)	29.39 (7.06)	28.44 (6.60)	31.02 (7.53)	<0.001
Years of service (years)	6.15 (5.52)	5.49 (5.19)	7.28 (5.88)	<0.001

Abbreviations: SD, standard deviation; NAFLD, non-alcoholic fatty liver disease; HDL-C, high-density lipoprotein cholesterol.

**Table 2 ijerph-19-16294-t002:** Multinomial logistic regression analysis of factors influencing NAFLD in nurses.

Predictors	NAFLD *n* = 260 (36.8%)	
	OR [95% CI]	*p*
Age (Years)	1.07 [1.00, 1.15]	0.044
Gender (Male vs. Female)	0.87 [0.31, 2.49]	0.798
Marital status (Living with vs. Not living with spouse or partner)	0.71 [0.38, 1.32]	0.281
Education		
Graduate school vs. Junior college	0.45 [0.15, 1.39]	0.166
University vs. Junior college	0.78 [0.50, 1.23]	0.283
Years of service	0.99 [0.90,1.08]	0.751
Drinking (Yes vs. No)	1.29 [0.83, 2.00]	0.262
Smoking (Yes vs. No)	0.70 [0.26, 1.86]	0.471
Shift work (Yes vs. No)	1.21 [0.68, 2.16]	0.525
BMI		
Obese vs. Normal	38.29 [14.74, 99.46]	<0.001
Overweight vs. Normal	4.12 [2.31, 7.32]	<0.001
Waist circumference (Exceeding standard vs. Normal)	2.27 [1.25, 4.14]	0.007
Hypertension (Yes vs. No)	0.35 [0.06, 2.13]	0.257
Fasting blood glucose (mg/dL)		
>126 vs. <100	3.91 [0.51, 30.01]	0.191
100–125 vs. <100	4.66 [1.42, 15.26]	0.011
Triglycerides (≥150 vs. <150) (mg/dL)	0.90 [0.33, 2.47]	0.832
Total cholesterol (≥200 vs. <200) (mg/dL)	1.25 [0.78, 2.01]	0.347
HDL-C (Overly low vs. Normal)	2.10 [1.11, 3.97]	0.022

Note: the reference category is ‘No NAFLD’, *n* = 446 (63.2%). Abbreviations: NAFLD, non-alcoholic fatty liver disease; BMI, body mass index; HDL-C, high-density lipoprotein cholesterol; OR, odds ratio; 95% CI, 95% confidence interval.

**Table 3 ijerph-19-16294-t003:** Multinomial logistic regression analysis of factors influencing NAFLD severity in nurses.

Predictors	NAFLD			
	Mild *n* = 188 (26.6%)		Moderate/Severe *n* = 72 (10.2%)	
	OR [95% CI]	*p*	OR [95% CI]	*p*
Age (Years)	1.08 [1.01, 1.16]	0.037	1.05 [0.94, 1.18]	0.398
Gender (Male vs. Female)	0.67 [0.21, 2.19]	0.510	6.42 [1.07, 38.70]	0.043
Marital status (Living with vs. Not living with spouse or partner)	0.71 [0.38, 1.32]	0.278	0.61 [0.20, 1.86]	0.380
Education				
Graduate school vs. Junior college	0.40 [0.12, 1.32]	0.132	0.47 [0.08, 2.79]	0.407
University vs. Junior college	0.79 [0.50, 1.24]	0.296	0.58 [0.26, 1.30]	0.186
Years of service	0.99 [0.90, 1.08]	0.760	1.02 [0.89, 1.17]	0.782
Drinking (Yes vs. No)	1.31 [0.84, 2.05]	0.230	0.99 [0.45, 2.18]	0.977
Smoking (Yes vs. No)	0.65 [0.24, 1.82]	0.416	1.67 [0.27, 10.34]	0.579
Shift work (Yes vs. No)	1.25 [0.70, 2.26]	0.453	0.88 [0.32, 2.41]	0.798
BMI				
Obese vs. Normal	23.30 [8.88, 61.10]	<0.001	409.63 [61.59, 2724.38]	<0.001
Overweight vs. Normal	3.89 [2.15, 7.04]	<0.001	13.50 [2.35, 77.50]	0.004
Waist circumference (Exceeding standard vs. Normal)	2.10 [1.14, 3.87]	0.017	9.33 [2.30, 37.98]	0.002
Hypertension (Yes vs. No)	0.16 [0.02, 1.52]	0.110	0.65 [0.07, 5.82]	0.697
Fasting blood glucose (mg/dL)				
>126 vs. <100	3.55 [0.43, 29.44]	0.240	2.98 [0.22, 40.11]	0.411
100–125 vs. <100	4.09 [1.19, 14.03]	0.025	11.74 [2.22, 62.27]	0.004
Triglycerides (≥150 vs. <150) (mg/dL)	0.57 [0.18, 1.80]	0.337	5.63 [1.35, 23.49]	0.018
Total cholesterol (≥200 vs. <200) (mg/dL)	1.29 [0.80, 2.07]	0.295	0.87 [0.36, 2.14]	0.767
HDL-C (Overly low vs. Normal)	2.01 [1.05, 3.85]	0.036	2.60 [1.03, 6.58]	0.044

Note: the reference category is ‘No NAFLD’, *n* = 446 (63.2%). Abbreviations: NAFLD, non-alcoholic fatty liver disease; BMI, body mass index; HDL-C, high-density lipoprotein cholesterol; OR, odds ratio; 95% CI, 95% confidence interval.

## Data Availability

The data analyzed during the current study are available from the corresponding author upon reasonable request.

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
