# Peer review of "Correlation between Component Factors of Non-Alcoholic Fatty Liver Disease and Metabolic Syndrome in Nurses: An Observational and Cross-Sectional Study"

_ijerph, 2022, doi:10.3390/ijerph192316294_

Round 1
Reviewer 1 Report
Chang et al. collected a total of 706 valid samples based on health checkup in nurses. This study found that the prevalence rate of NAFLD among nursed is 36.8%. Greater age, overweight and obese BMI, excessive waist circumference, fasting blood glucose between 100-125mg/dL and low HDL-C are at greater risk of NAFLD.
Some major point:
1. Authors should provide the standard/details of diagnoses for NAFLD by the hepatobiliary specialist. For example, drop lipid stage, inflammation stage, fibrosis......
2. The sample of male nurse is too small.
3. The total number of this study is small.
4. The classification of the nurse's department and length of service were not taken into consideration in this study.
5. Lifestyle, dirt habits and whether eat during shift work didn't included into this study.
6. What advice are there to prevent NAFLD in nurses?
Reviewer 2 Report
The present study is pertaining to understanding the correlation between non-alcoholic fatty liver disease (NAFLD) and metabolic syndrome in nurses and volunteers who were not included in the study those having drinking habits, hepatitis B or C, or incomplete data. The overall study is interesting and can be reconsidered after addressing following queries
Introduction is well written I would suggest adding a little about how much % of fat accumulates in the liver in NAFLD 10.4254/wjh.v7.i6.846 (Non-alcoholic fatty liver disease: The diagnosis and management)
In the study design did authors exclude the occasional drinkers and or having a history of drinking?
Did the authors confirm whether the nurses were taking any drugs and or any herbal supplements which may have the potential to induce steatosis?
I would recommend drawing one pictorial presentation of the study design where indicating BMI, glucose, and other parameters were included in the study design.
NAFLD is more in old age nurses than younger ones, which factors are more contributing to the development of NALFD in nurses.
Do authors find any difference; between male nurses being prone to NALFD or females?
If yes then why male are more prone than females? And justify the same in the discussion part.
Line 190 font size is different .?
Why didn’t the author go for LDL-Cholesterol?
Round 2
Reviewer 1 Report
Chang et al. collected a total of 706 valid samples based on health checkup in nurses. This study found that the prevalence rate of NAFLD among nursed is 36.8%. Greater age, overweight and obese BMI, excessive waist circumference, fasting blood glucose between 100-125mg/dL and low HDL-C are at greater risk of NAFLD. The revised manuscript has improved a lot. However, two mini points need to be modified.
Mini points:
1. Line 74 showed that this study excluded drinking habit. But, figure 1 showing 286 nurses drink. What are the categories of Drinking-Yes in Figure 1 base on line 80-81? Could author please clarify the drinking habits of samples used in this study based on line 80-81 as the study is talking about NAFLD? This sentence "Those that circled the last were eliminated (there was 1)" is not clear.
2. I still think there is a need to improve the English editing of this study and make it more professional. For example, the sentence in line 147-148 should be improved;
Author Response
In accordance with Reviewer #1’s suggestions:
- Line 74 showed that this study excluded drinking habit. But, figure 1 showing 286 nurses drink. What are the categories of Drinking-Yes in Figure 1 base on line 80-81? Could author please clarify the drinking habits of samples used in this study based on line 80-81 as the study is talking about NAFLD? This sentence "Those that circled the last were eliminated (there was 1)" is not clear.
Response:
We highly appreciate the reviewer’s reminder. We rewrote the following paragraph by incorporating Line 74 and Line 80-81: “Nurses in this study were assessed their drinking habits by clicking the right answer from the three choices: non-drinking, drinking alcohol occasionally (in social situations), and drinking alcohol every day. Those who chose the choice: ‘drinking alcohol every day’ were not allowed to participate in this study (Only one person clicked this choice and has been excluded from this study).” (see the Materials and Methods Section on Page 2 in line 74)
To clarify the situation, we’ve added a few words “including occasionally or only when entertaining” in Figure 1. (see the Figure 1)
- I still think there is a need to improve the English editing of this study and make it more professional. For example, the sentence in line 147-148 should be improved;
Response:
We highly appreciate the reviewer’s reminder. We’ve revised the two lines below: “The results from multinomial logistic regression analysis in Table 3 indicated that compared with nurses without NAFLD,” (see the Results Section on Page 5 in line 147-148)
We meanwhile requested a professional translation agency to double check the text. (see the Text)

Reviewer 2 Report
The manuscript can be accepted.
Author Response
In accordance with Reviewer #2’s suggestions:
- The manuscript can be accepted.
Response:
Thank you for this positive evaluation.
